# SEMI-SUPERVISED OUTLIER DETECTION USING GENERATIVE AND ADVERSARY FRAMEWORK

## ABSTRACT

In a conventional binary/multi-class classification task, the decision boundary is supported by data from two or more classes. However, in one-class classification task, only data from one class are available. To build a robust outlier detector using only data from the positive class, we propose a corrupted GAN (CorGAN), a deep convolutional Generative Adversary Network requiring no convergence during the training process. In the adversarial process of training the CorGAN, the Generator is supposed to generate outlier samples for the negative class, and the Discriminator is trained to distinguish training datasets (i.e., positive samples) from generated data from the Generator (i.e., negative samples). We also propose a lot of techniques to improve the performance of the built classifier (i.e., the Discriminator). The proposed model outperforms the traditional method PCA + PSVM (Schölkopf et al., 2000) and the solution based on Autoencoder (Thompson et al., 2002).

## 1 INTRODUCTION

(Hodge & Austin, 2004) addresses three fundamental approaches detecting outliers. The first approach is unsupervised clustering that identifies outliers without using any prior knowledge of the data. The second approach, supervised classification, requires labeled data from both positive class and negative class. The third addressed approach detects outliers using only data from the positive class via semi-supervised learning. Semi-supervised learning has gained increasing attention in recent years. One-class classification(OCC), as a typical semi-supervised learning technique, is applied to detect outliers using only positive examples from one class. The semi-supervised learning in this paper focuses on the OCC technique.

To motivate the importance of OCC, we first make an introduction to a classic application scenario. In industry, machine monitoring system is used everywhere to detect machine faults. A classifier should be constructed to detect when the machine behaves abnormally. Obviously, the training data for the positive class is easy to obtain by measuring the normal operations of the machine. However, only limited training data is available, even totally unavailable. In such case, a classifier should be built only on positive training data. This kind of task is known as OCC task. The name "one-class classification" originates from the paper Moya et al. (1993). Other researchers also present similar tasks with other terms such as Outlier Detection (Ritter & Gallegos, 1997), Novelty Detection (Bishop, 1994) or Concept Learning (Japkowicz, 1999). They are used interchangeably in this paper, even though they have specific meanings in other works. One-class classification can be used not only in machine monitoring task but also in many other domains, e.g. Text mining (Basu et al., 2004), Sentiment Analysis (Agarwal et al., 2015) and IT security (Lakhina et al., 2005).

Many solutions have been proposed to solve the one-class classification problem. However, almost none of them shows acceptable performance in high-dimensional space. Neural Network with deep architecture is well known for the ability to manipulate high-dimensional data. It achieves state-of-art results in speech recognition, visual object recognition, object detection and many other domains such as drug discovery and genomics (LeCun et al., 2015). This paper applies a neural network with deep architecture in outlier detection task. Generative adversary framework(GAN) is composed of a Generator $G$ that can be used to generate outliers and a Discriminator that can be trained as a binary classifier. The framework is a potential solution to detect outliers through generating counterexamples. Usually, the Nash equilibrium of the training process of GANs cannot be guaranteed

in practice. Our proposed model requires no convergence of the training process since the $G$ is used to generate only outliers instead of high-quality images that are from the distribution the training dataset. The proposed deep architecture solution is implemented, analyzed and compared to other methods.

The first section introduces the one-class classification problem and a potential solution with deep-architecture neral network. The second section presents the work related to one-class classification problems (i.e., semi-supervised outlier detection). Then, the two primary steps of our solution for one-class classification problem are described in the third section, namely, the training step to optimize model and the detecting step to make an inference. Next, the fourth section proposes a technique to break Nash equilibrium so that the $G$ of GAN can keep generating outliers. Besides, this section also proposes other techniques to improve. The fifth section shows experiments, analyzes the results and compares the performance with that of other methods. Finally, the last section concludes our work and describes future work that remains to be further researched.

## 2    RELATED WORK

Five approaches to solve OCC problem are summarized in (Pimentel et al., 2014). Probabilistic approach estimates the generative probability density function (pdf) of the data from the positive class. The boundaries of normality in the data space are defined by the resultant distribution together with a specified threshold, and an unseen sample is tested whether it comes from the same distribution or not. Thereinto, Gaussian Mixture Models (GMMs) (Lindsay et al., 1989; Bishop, 2006) and Kernel Density Estimators (Parzen, 1962; Vincent & Bengio, 2003; Bengio et al., 2006) have proven to be popular. This approach requires complete density estimation in the feature space. If the data in feature space are high dimensional, huge amounts of data are required to fit the model because of the curse of dimensionality. Only when the data from the target class are large enough can this kind of method perform well. Another well-known approach, Reconstruction-based approach, first train a model minimising the reconstruction error of training data with positive labels. Then, the trained model assigns an outlier score, the distance between the input representation vector and the output of the model, for each test example. (Markou & Singh, 2003) reviews lots of the neural network-based methods. Additionally, PCA can also detect outliers by comparing the example before and after transformation. The reconstruction error approach abandons some information with low variance during reconstruction. However, the abandoned low-variance information has proven to be most informative (Tax & Müller, 2003).

Additionally, Distance-based approach, e.g.  Nearest neighbour-based methods (Bay & Schwabacher, 2003; Breunig et al., 2000) and Clustering-based methods (Barbará et al., 2002; He et al., 2003), avoids estimating pdf explicitly, but it requires a well-defined distance/similarity measure, which is especially difficult in high-dimensional space. Another approach is domain-based, which creates the boundary based on the structure of normal data without considering the density of the positive class. One-class SVM (Schölkopf et al., 2000) and Support vector data description (SVDD) (Tax & Duin, 1999) are two basic ones. However, the choice of an appropriate kernel function is not easy, which determines the computational cost. Moreover, the hyperparameters that control the tightness of the boundary are also difficult to select. Lastly, Information-theoretic approach tries to distinguish normal data and outliers by computing information content of dataset using information measure. Similarly, the selection of appropriate information-theoretic measure is challenging.

The approaches described above learn from available positive samples only. Approaches that learn from both target samples and artificial outliers are also researched. (Hempstalk et al., 2008; Fan et al., 2004) generate outlier with a predefined distribution. The strong assumptions about the outlier data distribution in these approaches may be violated in real datasets (Abe et al., 2006). (Tax & Duin, 2001) proposes a method for generating artificial outliers, uniformly distributed in a hypersphere. However, in high-dimensional data space, their proposed technique is not feasible anymore because it is tough to get a confident estimate of the target volume due to the large difference in volume of the target and outlier class. (Bánhalmi et al., 2007) extends dataset by generating outlier examples distributed around the positive class. The approach first finds boundary points explicitly using SVM, which is computationally expensive. Then it generates negative examples only around positive class using a distance measure, which causes infeasibility in high-dimensional space. Our

proposed CorGAN generates negative examples including both ones around the positive class and ones far from the positive class. Moreover, the model requires no explicit distance measure and does not need to find boundary points explicitly.

Neural networks with deep architecture have already been used in OCC task, but mostly in Reconstruction error approaches (Markou & Singh, 2003). To our knowledge, our proposed CorGAN is the first work to generate outliers for OCC via deep architecture (i.e., Generative Adversary Network). A variant of the GAN framework (CatGAN) is applied to solve multi-class classification task in unsupervised or semi-supervised fashion (Springenberg, 2015). (Odena, 2016) does a further research about semi-supervised learning using GANs. (Schlegl et al., 2017) proposes AnoGAN to apply GAN in Anomaly Detection, which requires the Nash-equilibrium at the end of the training process. Nevertheless, all variants of GAN and the original one are known for its unstable training process.

## 3 OUTLIER DETECTION USING CORGAN

The proposed model and the improved techniques can be generalized to various kinds of data. To show the performance in high-dimensional space, we illustrate our model on image data. The proposed parametric method is composed of two steps:

1. Training Step: Training the CorGAN with the improved techniques;
2. Inference Step: Detecting outliers using the resulting $D$ of the trained CorGAN.

### 3.1 GENERATIVE ADVERSARY NETWORK

Generative Adversary Network(GAN) is a framework for training generative models via an adversarial process (Goodfellow et al., 2014). The framework consists of two components, a generative model (Generator $G$) and a discriminative model (Discriminator $D$). The $G$ aims to capture the data distribution. The $D$ estimates the probability that a sample came from the training data rather than the Generator. This framework corresponds to a minimax two-player game. In the training procedure, the $D$ is trained to distinguish samples in training datasets from generated samples by assigning a high probability to the former and a low probability to the latter. Contrarily, the objective of $G$ is to maximize the probability of $D$ making a mistake. After the Nash-equilibrium of the training process, the output probability of the $D$ is always 0.5. In case of the convergence, the $G$ is capable of generating realistic images that have same/similar distribution as in training dataset, and the $D$ cannot make right discrimination anymore. The biggest advantage of this framework is that no Markov chains or unrolled approximate inference networks are required in the training and sampling process.

### 3.2 STEP1: TRAINING THE CORGAN

Architectures of the Generator and the Discriminator are neural networks, such as Multilayer Perceptron, Deep Convolutional Neural Network (LeCun et al., 1989), Convolutional Neural Network Cascade (Springenberg, 2015) and Recurrent Neural Network (Rumelhart et al., 1988). The Back-Propagation algorithm can be used to train both the generative model and the discriminative model. The architecture applied in the proposed CorGAN is shown in Figure 1.

The $G$ generally starts from prior distribution $p_z(\boldsymbol{z})$ (input noise variable $\boldsymbol{z}$). In the case of convergent GANs, the $G$ maps the prior distribution to the training data distribution $p_{inlier}(\boldsymbol{x})$. The $G$ of CorGAN is used to generate outlier examples. Hence, it is supposed to map the prior distribution to outlier data distribution $G(\boldsymbol{z}; \theta_g)$ instead of the training data distribution. As usual, the $D$ maps the input (i.e. the training data or the generated samples) to a single scalar, which represents the probability that the input came from training datasets instead of the $G$. The target value of the $D$ is $a_t = 1$ for the input data from training dataset and $a_o = 0$ for the input data generated by the $G$. The $D$ as a binary classifier is trained to minimize the cost V(D):

$$\min_D V(D) = \mathbb{E}_{\boldsymbol{z} \sim p_z(\boldsymbol{z})} \log(D(G(\boldsymbol{z})) - a_o) + \mathbb{E}_{\boldsymbol{x} \sim p_{inlier}(\boldsymbol{x})} \log(a_t - D(\boldsymbol{x})) \tag{1}$$

The objective of the $G$ of the CorGAN is to fool the D, but not necessarily maximise the probability D making a mistake. The new target value is $a_{new} \in [0, 1]$ (see section 4.2). The $G$ of the CorGAN

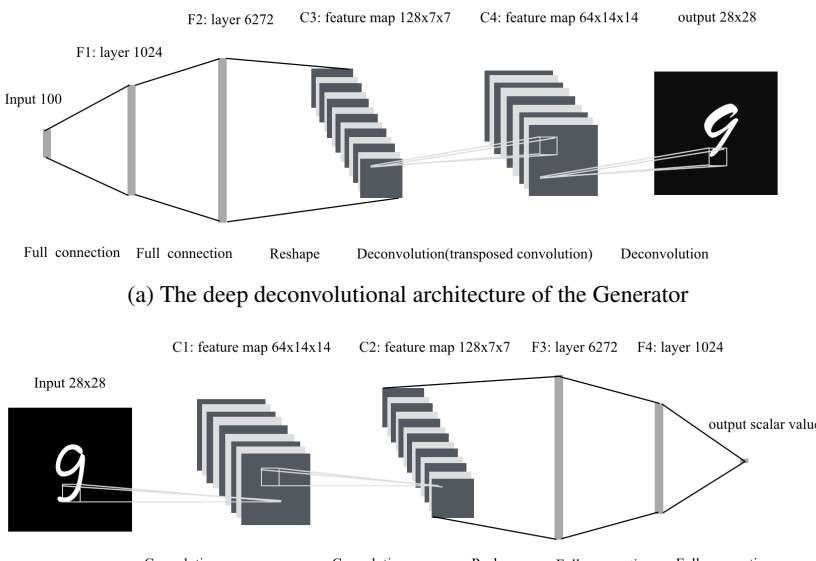

(a) The deep deconvolutional architecture of the Generator

(b) The deep convolutional architecture of the Discriminator

Figure 1: The basic architecture of the CorGAN

is trained to minimise the cost U(G):

$$\min_G U(G) = \mathbb{E}_{\boldsymbol{z} \sim p_z(\boldsymbol{z})} \log(|a_{new} - D(G(\boldsymbol{z}))|) \tag{2}$$

The CorGAN model is updated via back-propagation algorithm. If the $D$ is overly optimised without updating the $G$, it will result in overfitting problem. The $D$ and the $G$ will be updated simultaneously or alternately to avoid the problem, e.g. k steps of optimizing the $D$ and one step of optimizing the $G$. The traditional GANs reach Nash equilibrium after several training epochs. The new objective of the $G$ of CorGAN breaks Nash equilibrium of the training process, which causes that the $G$ can keep generating outlier samples.

The inlier data is taken as training data in the CorGAN. In the adversarial process of training Cor-GAN, the $G$ is supposed to generate outlier samples for the negative class. The $D$ is trained to assign a high probability value to data from training datasets (i.e., the positive class) and a small probability value to generated data from the G (i.e., the negative class). The generated outliers not only distribute around the positive class but also cover feature space far away from the positive class. In order that the $G$ can map a prior distribution to a huge data space except for the positive class, we proposed a lot of improved techniques (section 4).

### 3.3 STEP2: DETECTING OUTLIERS USING DISCRIMINATOR

In the inference step, the resulting $D$ outputs a relatively high probability for data subjective to the distribution $p_{inlier}$ and a relatively low probability for data not from the distribution $p_{inlier}$. That is to say that, if the output is a low probability in the outlier-detecting process, the input is predicted as an outlier. What is a low probability? So, we need a probability threshold to decide whether an output probability is high or low. The output of the sigmoid activation function of the last layer is a scalar value in the interval (0, 1), we can intuitively set $t$ as the threshold. In that case, the input is an outlier, if the output from the $D$ is small than $t$, otherwise an inlier.

The one-class classification task is an extreme case of the imbalanced training. The optimal value of the threshold $t$ is not 0.5. It mainly depends on how the model is trained and the concrete application scenario. If the model is trained by specifying a new objective for the $G$ (like in CorGAN), the $D$ model learns distribution from training datasets for a long time. However, the $D$ is trained with data from a more extensive outlier distribution using the same time. The resulting $D$ will present a

relatively higher probability for data that follow the same distribution as the training data (i.e., for inliers). So, the threshold $t$ with a value higher than 0.5 shows a better performance. We do not evaluate the $D$ on a single user-specified threshold.

One-class classification, also called Outlier Detection, can be evaluated with F1-score, which is harmonic mean of precision and recall. The accepted fraction of the positive class $f_{T+}$ and the rejected fraction of the negative class $f_{O-}$ are both together also as a popular measure for OCC. However, the score of those measures strongly depends on the specified threshold. To justify our model objectively, the performance of the $D$ in this paper will be evaluated with Receiver operating characteristic curve (ROC) and Area under the ROC curve (AUC). The robustness of the built $D$ will be tested on various datasets.

## 4 IMPROVED TECHNIQUES FOR GAN IN OCC

If the training process reaches Nash equilibrium, the $G$ is able to generate examples following the distribution $p_{inlier}$ (see figure 2), and the output probability of the $D$ is always 0.5 for inliers and an unexpected value for outliers. It is difficult to distinguish outliers from inliers via a threshold. Our proposed corrupted generative adversary network (CorGAN) is a GAN without convergence. To avoid the Nash equilibrium that the training process can reach, we propose several techniques to break the convergence and build a robust outlier identifier. Thereinto, specifying a new objective for the $G$ is a basic one to keep it generating outlier samples, and other optional techniques further improve the performance of the model.

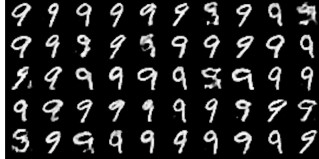 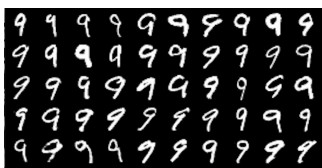

    (a) the generated images from Generator         (b) the images in training dataset

Figure 2: Comparison between the generated data and the training data: The images of handwritten digit nine are training data. After several training epochs, the generated images and the training data are visualised in the figure.

### 4.1 EARLY STOPPING

In early training epochs (i.e., before convergence), the $G$ has no ability to generate data that follows the distribution $p_{inlier}$. Meanwhile, the $D$ is trained with the training data with positive labels and the generated data with negative labels. Distributions from $G$ are different from the distribution of training datasets before convergence. The $D$ recognizes the distribution of training datasets by presenting a high probability. Early Stopping before convergence can obtain a well-behaved Discriminator.

In term of implementation of this technique, we do not explicitly stop the training at a particular epoch, but always save the best model. Similar to the model selection, we take the best Discriminator as the final classifier, which appears definitely before the convergence of the training process. The performance of the $D$ is tested regularly during the training process. The score Area Under the Curve of $f_{T+}$ (inlier accepted fraction), called positively biased AUC (see figure 3) is used to evaluate the performance of the $D$. The $D$ saved with best biased AUC score shows not optimal but near-optimal performance on the test datasets. The objective of Early Stopping is defined as follows:

$$\max_{D} AUC_{biased} = \int_0^1 f_{T+}(t)dt \tag{3}$$

where $t$ is the threshold and $f_{T+}(t)$ is inlier accepted fraction of the Discriminator given the specific threshold $t$.

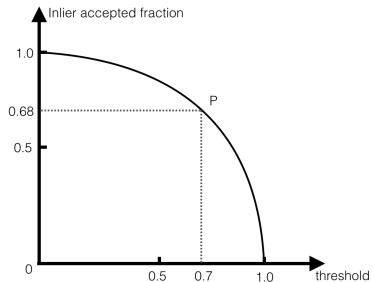

Figure 3: Area Under the Curve of inlier accepted fraction: The figure describes the relationship between the inlier accepted fraction and the specified threshold. Given the specified threshold *0.7*, the point *P* in the curve corresponds to the accepted fraction of inliers *0.68*. Since no outlier is available, the area under this curve (positively biased AUC) is a good measure to select the near optimal model.

| new target value $a_{new}$ |
| --- |
| $a_{new} = 1$: The objective of *G* is the exact same as that of the convergent GAN (Goodfellow et al., 2014). The training will converge. |
| $a_{new} \in (\sim 0.9, 1)$: The such adjustment of the objective of the *G* is proposed in (Salimans et al., 2016) to improve the training process of GANs. The training process will converge as well. |
| $a_{new} \in (\sim 0.5, \sim 0.9)$: The *G* will generate data far from the distribution $p_{inlier}$ at the beginning of the training phase because of the random initialization. After several training epochs, it will generate data that distribute around the positive class. The tighter the boundary is, the larger space the generated data cover. The value 0.9 results in most tight boundary. |
| $a_{new} \in [0, \sim 0.5)$: The *G* has similar objective to that of the *D*. It will tend to generate data, from which the *D* can easily distinguish the training data. That is to say that all the generated data distribute far from the distribution $p_{inlier}$. |

Table 1: The behavior of the *G* and the performance of the *D* are presented in case of different new target values.

## 4.2 Specifying a New Objective for the Generator

Even though Early Stopping avoids the problem the convergence causes, GAN can only be trained with a limited number of epochs. Hence, Early Stopping can only guarantee a high inlier accepted fraction $f_{T+}$, but not necessarily high outlier rejected fraction $f_{O-}$ because the D is only trained with a certain number of generated samples (i.e., outliers). To build a robust outlier identifier against as many kinds of outlier distributions as possible, we should train the *D* with as many generated samples as possible, which have different distribution from the distribution $p_{inlier}$.

We can explicitly break Nash equilibrium by specifying a new objective for *G*. Without modification, the objective of *G* is to maximise the probability of the *D* making a mistake. We propose a new objective for *G* :

$$\min_G U(G) = \mathbb{E}_{\boldsymbol{z} \sim p_z(\boldsymbol{z})} \log(|0.9 - D(G(\boldsymbol{z}))|) \tag{4}$$

Instead of maximising the probability that *D* makes a mistake, the new objective is that the *D* makes a mistake with a certain probability. The new target value used to calculate the cost for updating the G is $a_{new} = 0.9$. The choice of the value $a_{new}$ is justified in the table 1. In case of $a_{new} = 0.9$, the *G* explores the largest space, and the built *D* will show robust performance.

### 4.3 ATTACHING MORE IMPORTANCE TO GENERATED DATA

The cost of the $D$ consists of two parts. These two parts are caused respectively by the training data and the generated data. Generally, the two parts are simply added together as the total cost for updating the parameters of the $D$. That is to say that the training data and the generated data are treated with the same importance. They can be treated differently by assigning a weight to one of them to broaden the search space of parameters. The objective of the $D$ is defined as follows:

$$\min_D V(D) = \mathbb{E}_{\boldsymbol{z} \sim p_z(\boldsymbol{z})} \log(D(G(\boldsymbol{z})) - a_o) + w * \mathbb{E}_{\boldsymbol{x} \sim p_{inlier}(\boldsymbol{x})} \log(a_t - D(\boldsymbol{x})) \tag{5}$$

, where $w \in (0, 1)$ is a hyperparameter. The value of $w$ can be selected by validation process with positively biased AUC score. While the outlier distributions are various and difficult to recover all of them, the inlier distribution is rather simple and easy to learn. During the training process, the cost that generated data caused should be reduced as far as possible by updating parameters of the $D$. In other words, the generated data should be attached more importance by specifying the value of weight. Compared to the general case that the two parts of cost are not treated differently, this method shows a better performance on the test datasets whose distributions are far from the training dataset.

### 4.4 COMBINING PREVIOUSLY GENERATED DATA

Compared to the method of Early Stopping, the method of specifying a new objective for G presents a better performance, because the new objective trains $D$ with arbitrarily more generated data that are not from the distribution $p_{inlier}$. With the new specified objective, the training procedure does not converge, and the $G$ is able to keep generating outliers. The $D$ can be trained with arbitrarily many generated distributions. However, the space of distribution learned by $D$ is limited to a great extent. On the one hand, the generated distribution always stays near the positive class after several training epochs. On the other hand, the $D$ can forget the previously learned distributions because of the limited capacity.

In this subsection, we proposed a technique to broaden the learned distributions. The performance of the $D$ can be improved by being regularly trained with previously generated data. We can train the CorGAN with mini batches (batch size $s$) that combine the data generated recently and previously. The combined training data can avoid that the $D$ forgets the learned distribution to some degree.

There exist a large amount of generated data in the training procedure. Which ones should be chosen to train $D$ and prevent it forgetting the previously generated distributions? Because the generated data can be arbitrarily many, it is inadvisable and impossible to save all of them. In this case, the generated data can be treated as stream data $(X_1, X_2, \ldots, X_t)$. We apply a Reservoir Sampling Algorithm (Vitter, 1985) to sample previously generated images. This algorithm samples examples from the stream data with the same probability (see equation 6) and specifies a reservoir $R$ to save the sampled examples.

$$P(X_i \in R) = \frac{1}{t - (s/2)} \tag{6}$$

, where $i \in [1, t - (s/2)]$. The mini batches that are composed of newly generated examples and the sampled examples saved in a reservoir is used to train $D$. The mini batch $B$ at the timestamp $t$ is defined as follows:

$$B = \{R, X_{t-(s/2)+1}, X_{t-(s/2)+2}, \ldots, X_t\} \tag{7}$$

, where $R$ is the reservoir. The objective of the $D$ remains unchanged in the equation 1. The resultant $D$ can identify not only recently generated outliers but also previous ones.

## 5 EXPERIMENTS AND ANALYSIS

In this section, we justify our proposed model and improved techniques with experiments. To demonstrate the robust performance of the built classifier, we evaluate the $D$ on various outlier datasets. We describe the experiment settings of our models and the models to be compared. The experiment results, followed by a strong discussion, are presented in this section.

| Datasets: | Source of images: | The Number of images: |
|---|---|---|
| Training dataset | digit of 9 in MNIST | 4967 |
| Validation dataset | digit of 9 in MNIST | 900 |
| Test dataset | Inliers: digit of 9 in MNIST | 900 |
| | 1.Outliers: digits of 0-8 in MNIST | 900 |
| | 2.Outliers: CIFAR10 | 900 |
| | 3.Outliers: Images composed of noise | 900 |

Table 2: Training -, validation - and test datasets of experiments setting.

## 5.1 DATASETS AND EVALUATION:

Three datasets are used in the experiments, namely, MNIST (LeCun et al., 1998), CIFAR10 (Krizhevsky, 2009) and an artificial noise image dataset. The image size in MINIST is (28, 28). The size of CIFAR10 images is cropped into (28, 28) by removing pixels along the sides. Especially, we specify a dataset composed of three group of noise images with the same size (28, 28). The values of their pixels are respectively subject to uniform distribution, Gaussian distribution and random values. The table 2 lists training dataset, validation dataset and test datasets. The performance of various approaches will be evaluated and compared with Receiver Operating Characteristic (ROC) and the Area Under the ROC Curve (AUC).

## 5.2 EXPERIMENTS SETTING:

**PCA+PSVM:** PCA is used to reduce the dimensionality of the high-dimensional data (i.e., images). The number of components K is set such that 95% of the variance is retained (K=111). One-class SVM proposed in (Schölkopf et al., 2000) is plane-based, called PSVM. To identify outliers in the feature space, PSVM tries to find a hyperplane that best separates the data from the origin. RBF kernel is used in this experiment. Other settings are defaults in sklearn.svm.OneClassSVM (Pedregosa et al., 2011).

**Autoencoder:** Autoencoder detects outliers by computing reconstruction error and compares it with a specified threshold. The threshold is based on the difference between the inputs and outputs for the training data. If the reconstruction error for a test sample is larger than the threshold, then the sample is identified as an outlier, otherwise as inlier. To justify our proposal, we compare our model to convolutional autoencoder. The encoder has the same architecture as the Discriminator in CorGAN except for output layer. The decoder also has a same architecture as the Generator in CorGAN. The model is regularised with weight decay $\lambda$ =0.01. The parameters are updated with SGD optimisation algorithm, *minibatch*=128 and learning rate *lr*=0.1. The cost function is the cross-entropy function. The model is trained for 30 epochs without pretraining.

**CorGAN:** The basic architecture of CorGAN, as well as the number of its layers and units, is shown in figure 1. We propose a lot of improved techniques. Since its combinations are numerous, we justify only three main models. The first model is a basic one, CorGAN = GAN with early stopping technique and a new objective for the G (see section 4.2). The new target value $a_{new}$ is set manually to 0.9 for the G. The G is regularised with weight decay $\lambda = 0.1$. The optimisation algorithm is Adam, *minibatch* = 128 and learning rate *lr* = 0.0002. No pretraining is performed. The second model to be justified is based on the first one, CorGAN[2] = CorGAN + Attaching more importance to generated images (see section 4.3). The weight is set to 0.5 manually. The third illustrated model is also based on the first one, CorGAN[3] = CorGAN + Combining previously generated images (see section 4.4). The minibatch size is composed of 64 images sampled from previous training epoch and 64 newly generated images.

## 5.3 RESULTS AND ANALYSIS:

The results of the experiments are shown in the figure 4 and the table 3. The outlier distribution of the handwritten digits images of the numbers (0-8) is relatively close to the inlier distribution of the number 9. Hence, all the approaches show the worse AUC scores on the first test dataset. The

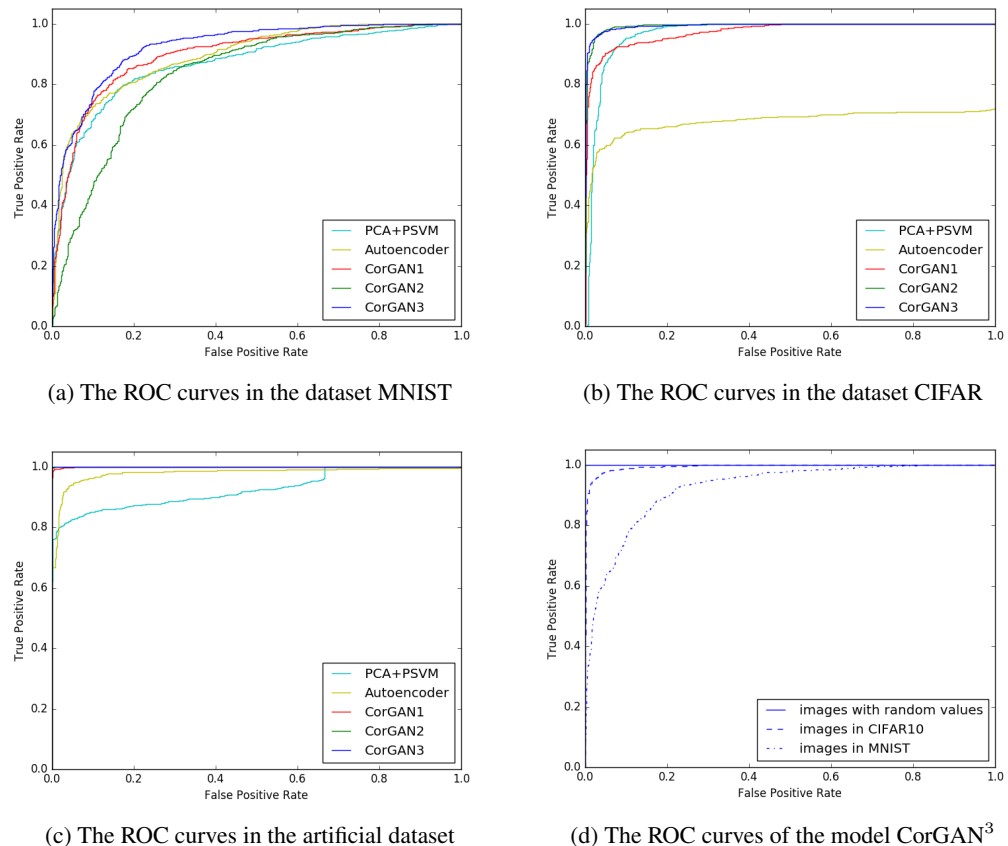

(a) The ROC curves in the dataset MNIST

(b) The ROC curves in the dataset CIFAR

(c) The ROC curves in the artificial dataset

(d) The ROC curves of the model CorGAN$^3$

Figure 4: The figures show the ROC curves of all models on three differenct test datasets. The area under the ROC curve represents the overall performance of a one-class classifier. The model CorGAN$^3$ shows robust performance on all the three datasets.

| | AUC score: | | |
| | MNIST(9) | | |
| | MNIST(0-8) | CIFAR10 | Noise |
|---|---|---|---|
| PCA + PSVM | 0.8623 | 0.9720 | 0.9302 |
| Autoencoder | 0.8943 | 0.6785 | 0.9704 |
| CorGAN | 0.8974 | 0.9739 | 0.9995 |
| CorGAN$^2$ | 0.8343 | 0.9937 | **0.9999** |
| CorGAN$^3$ | **0.9253** | **0.9943** | **0.9999** |

Table 3: The AUC socres of various models are shown in the table. All the models are tested in three datasets: MNIST(9) + MNIST(0-8), MNIST(9) + CIFAR10, MNIST(9) + Noise. Within MNIST(9) images are inliers, and other images are outliers. CorGAN, CorGAN$^2$ and CorGAN$^3$ are described in section 5.2.

PCA+PSVM approach shows the better score on the second test dataset than on the noise test dataset. The traditional approach is not robust enough for noise outliers. The solutions based on neural networks often show a better performance against noise data because of the random initialization of its parameters. Especially, our proposed solution based on GAN framework, in which Generator generates many noise examples. The convolutional autoencoder can reconstruct natural images well by detects edges, corners and objects. Therefore, the convolutional autoencoder shows the poor score on natural images. Our proposed solution classifies test examples without reconstruction process, which shows robust performance against outlier natural images as well as noise images.

Compared to CorGAN, CorGAN[2] attaches more importance to generated images, which makes classifier more robust again the outliers whose distribution is far from inlier distribution. In consequence, CorGAN[2] shows the low score on the first dataset, in which the distributions of inliers and outliers are relatively close. In the model CorGAN[3], the outlier examples generated previously are combined with newly generated examples to train the Discriminator. In this way, the Discriminator learned a large space of outlier distribution. The model CorGAN[3] shows the best scores on various test datasets. The robust CorGAN[3] learns a tight boundary in high-dimensional space. The farther the outlier distribution is from the inlier distribution $p_{inlier}$, the better score it shows (see figure 4d).

## 6 CONCLUSION AND FUTURE WORK

In this paper, we present a solution to solve one-class classification problem based on GAN framework and successfully apply the Discriminator of the framework to detect outliers. We illuminate a few techniques to improve the performance and verify the proposed techniques with experiments. First, we choose the near optimal model to detect outliers by saving a better model during the training procedure. Then we specify a new objective for the $G$ so that it can keep generating outliers. Attaching more importance to generated images can further improve the performance of the $D$. To prevent the $D$ forgetting the previously generated outliers, we combine previously generated outliers from the Generator to train the outlier identifier. These techniques show comparable AUC scores.

In future work, We can further vary the generated outliers to train $D$. We can specify multiple Generators in the generative adversary framework. The mini batch can combine the data generated by different Generators, which have different objectives, e.g. the different probabilities of D making a mistake. To further explore more generated distribution used to train the $D$, we can even combine CorGAN with other generative models. Similarly, we must also change the objective of them to fit our goal, since the other generative models are also supposed to generate outliers.

All the proposals in this paper do not leverage distance information(KLD) between distributions within a batch both in the training process and detecting process. Another topic worth studying is the clustering-based method to detect outlier using $D$ of GAN. One potential method of leveraging distance information is to model the closeness between examples in a mini-batch. The modeling process is described in Minibatch Discrimination (Salimans et al., 2016), an improved technique for training GANs.

Regarding the task of detecting of outlier images, we will try to identify more fine attributes of images. For instance, the built outlier identifier should be able to distinguish images taken under different illumination as well as different viewpoints, which describe the same object. Furthermore, we can take images of a group of objects as inliers. We will build a one-class classifier to make a decision whether the object described by the given image comes from the group.

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
