# OpenReview forum: "Semi-supervised Outlier Detection using Generative And Adversary Framework"
_ICLR.cc/2018/Conference — Reject_

### Official Review · AnonReviewer2 · 2017-11-26
**Interesting idea, but paper and experiments need revision**

**Rating:** 4
**Confidence:** 4

**Review:**

The idea of the paper is to use a GAN-like training to learn a novelty detection approach. In contrast to traditional GANs, this approach does not aim at convergence, where the generator has nicely learned to fool the discriminator with examples from the same data distribution. The goal is to build up a series of generators that sample examples close the data distribution boundary but are regarded as outliers. To establish such a behavior, the authors propose early stopping as well as other heuristics.

I like the idea of the paper, however, this paper needs a revision in various aspects, which I simply list in the following:
* The authors do not compare with a lot of the state-of-the-art in outlier detection and the obvious baselines: SVDD/OneClassSVM without PCA, Gaussian Mixture Model, KNFST, Kernel Density Estimation, etc
* The model selection using the AUC of "inlier accepted fraction" is not well motivated in my opinion. This model selection criterion basically leads too a probability distribution with rather steep borders and indirectly prevents the outlier to be too far away from the positive data. The latter is important for the GAN-like training.
* The experiments are not sufficient: Especially for multi-class classification tasks, it is easy to sample various experimental setups for outlier detection. This allows for robust performance comparison.
* With the imbalanced training as described in the paper, it is quite natural that the confidence threshold for the classification decision needs to be adapted (not equal to 0.5)
* There are quite a few heuristic tricks in the paper and some of them are not well motivated and analyzed (such as the discriminator training from multiple generators)
* A cross-entropy loss for the autoencoder does not make much sense in my opinion (?)


Minor comments:
* Citations should be fixed (use citep to enclose them in ())
* The term "AI-related task" sounds a bit too broad
* The authors could skip the paragraph in the beginning of page 5 on the AUC performance. AUC is a standard choice for evaluation in outlier detection.
* Where is Table 1?
* There are quite a lot of typos.

*After revision statement*
I thank the authors for their revision, but I keep my rating. The clarity of the paper has improved but the experimental evaluation is lacking realistic datasets and further simple baselines (as also stated by the other reviewers)

---

> ### Author Response · Authors · 2017-12-17
> **RE: Interesting idea, but paper and experiments need revision**
>
> Thank you for your comments,  our response to your questions are as follows:
> * The authors do not compare with a lot of the state-of-the-art in outlier detection and the obvious baselines: SVDD/OneClassSVM without PCA, Gaussian Mixture Model, KNFST, Kernel Density Estimation, etc
>
> 1) SVDD/OneClassSVM without PCA
> Using GAN Framework, we aim to build an one-class classifier that demonstrates robust performance in high-dimensional. There are so many irrelevant attributes in high dimensional (e.g. Image data). Therefore, it makes more sense to apply PCA to the data. Without PCA, the OneClassSVM shows a bad performance, AUC score = 0.6830.
>
> 2) Gaussian Mixture Model, KNFST, Kernel Density Estimation
> All the methods you named belong to the Density-based approach. The approach is known for the curse of dimensionality. It requires a very large number of training samples in high-dimensional space. We only compare our methods with the OneClassSVM method and the Autoencoder-based method. To our knowledge, They are respectively the state-of-the-arts of the Boundary-based approach and the Reconstruction Error-based approach.
>
> * The model selection using the AUC of "inlier accepted fraction" is not well motivated in my opinion. This model selection criterion basically leads too a probability distribution with rather steep borders and indirectly prevents the outlier to be too far away from the positive data. The latter is important for the GAN-like training.
>
> We propose a measure, called positively biased AUC, to select the model. If the generated outliers are far away from the positive class, it is easier for the Discriminator to distinguish the inliers and outliers. The Discriminator can show a good score. In my opinion, contrary to what you said, the measure indirectly brings the outlier far away from the positive data.
>
> The Generator generates outlier far away from the positive class at the beginning of training because of random initialisation. With the training going on, the new target of the Generator will drive it to generate negative data around the positive class. The generated data covers a large space. The indirect influence of the measure is too small, compared to the impact of the Generator on the generated outliers. In addition, the measure is not supposed to select the optimal model, but near optimal. That is why we call it positively biased AUC.
>
> * The experiments are not sufficient: Especially for multi-class classification tasks, it is easy to sample various experimental setups for outlier detection. This allows for robust performance comparison.
>
> Every multi-class classification task can be transferred into multiple binary classification tasks. We can demonstrate the performance of the binary classifier on multi-class classification tasks. But it is not a good idea for the one-class classifier. The one-class classifier should be robust against not only other classes in the same dataset but also any other samples in other datasets, even noise.  I hope I understand your question correctly.
>
> * With the imbalanced training as described in the paper, it is quite natural that the confidence threshold for the classification decision needs to be adapted (not equal to 0.5).
>
> Our task setting is one-class classification, in which we have no any outlier. It is an extreme fall of the imbalanced training. You are right, the best threshold is by no mean 0.5. But, in our paper, we do not use any confidence threshold. We evaluate the performance of the outlier detector with AUC score. The score takes all the confidence thresholds into consideration and gives an overall performance of the detector.
>
> * There are quite a few heuristic tricks in the paper and some of them are not well motivated and analyzed (such as the discriminator training from multiple generators)
>
> We only justify three ideas, namely, Attaching more importance to generated data, Specifying a new objective for the Generator and Combining the previously generated outliers. They are CorGAN, CorGAN2 and CorGAN3 in the experiments respectively. The proposed Early Stopping is applied to all of them and it serves the model selection. The two ideas in section future work are not justified at all in our paper. That is why we put them in the future work section.
>
> * A cross-entropy loss for the autoencoder does not make much sense in my opinion (?)
>
> Both MSE and Cross-Entropy make sense for the Autoencoder. See the link http://deeplearning.net/tutorial/dA.html#daa
>
> Thank you for the minor comments. We definitely should fix them.  By the way, the Table 1 is below the Figure 3 on the page 6.
>
> If we misunderstand some questions or you have any other question,  just let us know. We are very glad to further discuss with you.

---

### Official Review · AnonReviewer3 · 2017-11-27
**Questionable formulation with insufficient experiments**

**Rating:** 4
**Confidence:** 3

**Review:**

The idea of using GANs for outlier detection is interesting and the problem is relevant. However, I have the following concerns about the quality and the significance:
- The proposed formulation in Equation (2) is questionable. The authors say that this is used to generate outliers, and since it will generate inliers when convergence, the authors propose the technique of early stopping in Section 4.1 to avoid convergence. However, then what is learned though the proposed formulation? Since this approach is not straightforward, more theoretical analysis of the proposed method is desirable.
- In addition to the above point, I guess the expectation is needed as the original formulation of GAN. Otherwise the proposed formulation does not make sense as it receives only specific data points and how to accumulate objective values across data points is not defined.
- In experiments, although the authors say "lots of datasets are used", only two datasets are used, which is not enough to examine the performance of outlier detection methods. Moreover, outliers are artificially generated in these datasets, hence there is no evaluation on pure real-world datasets. To achieve the better quality of the paper, I recommend to add more real-world datasets in experiments.
- As discussed in Section 2, there are already many outlier detection methods, such as distance-based outlier detection methods, but they are not compared in experiments.
  Although the authors argue that distance-based outlier detection methods do not work well for high-dimensional data, this is not always correct.
  Please see the paper:
  -- Zimek, A., Schubert, E., Kriegel, H.-P., A survey on unsupervised outlier detection in high-dimensional numerical data, Statistical Analysis and Data Mining (2012)
  This paper shows that the performance gets even better for higher dimensional data if each feature is relevant.
  I recommend to add some distance-based outlier detection methods as baselines in experiments.
- Since parameter tuning by cross validation cannot be used due to missing information of outliers, it is important to examine the sensitivity of the proposed method with respect to changes in its parameters (a_new, lambda, and others). Otherwise in practice how to set these parameters to get better results is not obvious.

* The clarity of this paper is not high as the proposed method is not well explained. In particular, please mathematically formulate each proposed technique in Section 4.

* Since the proposed formulation is not convincing due to the above reasons and experimental evaluation is not thorough, the originality is not high.

Minor comments:
- P.1, L.5 in the third paragraph: architexture -> architecture
- What does "Cor" of CorGAN mean?

AFTER REVISION
Thank you to the authors for their response and revision. Although the paper has been improved, I keep my rating due to the insufficient experimental evaluation.

---

> ### Author Response · Authors · 2017-12-16
> **RE: Questionable formulation with insufficient experiments**
>
> Thank you for your comments,  the answers to your questions are following:
>  - The proposed formulation in Equation (2) is questionable. ... ... Since this approach is not straightforward, more theoretical analysis of the proposed method is desirable. What does "Cor" of CorGAN mean?
>
> The main idea of the paper is to corrupt the GAN with proposed technique so that the GAN does not converge. In the case of the corrupted GAN, the Generator is able to keep generating outliers. That is why we call it CorGAN (corrupted GAN).
>
> One of the techniques is Early Stopping. If we take the Discriminator before the convergence, all the generated samples used to train D are outliers.
>
> Another technique is specifying a new objective for the Generator. Alpha_new in the equation 2 is a variable. It is not necessary 1 as in the original GAN. If its value is from the interval (0, 0.9), the GAN is not capable of getting converged. Without convergence, the Generator will generate only outliers. See more details in response to the first review comment.
>
> - In addition to the above point, ... ... how to accumulate objective values across data points is not defined.
>
> Our built Generator is supposed to generate data far from the positive class and also data around the positive class. That is to say that the Generator should explore as a large space as possible. We care for the distribution of all the generated samples. We are not interested in where the training process ends exactly. So the expectation of the formulation is not that relevant. That is also a reason why we propose combining previously generated outliers to training the Discriminator. The accumulation of the generated outliers is implemented by the Reservoir Sampling Algorithm (see the last paragraph in the section 4.4 in our paper).
>
> - In experiments, although the authors say "lots of datasets are used", ... ... To achieve the better quality of the paper, I recommend to add more real-world datasets in experiments.
>
> We aim to build a robust outlier detector against any kind of outliers. Actually, we build just one outlier detector and generate outliers with help of only one training dataset, i.e. digit of 9 in MNIST. In order to test the robustness of the outlier detector, we use three outlier test datasets. The first one is digits of 0-8 in MNIST to test how tight the boundary is. The second one is a real world image dataset. We test the performance of the detector trained on MNIST on a real-world dataset (CIFAR10). In addition, we also demonstrate the robustness of the detector against not only real-world images but also noise. Therefore, we artificially create images. The values of their pixels are random or subject to various distributions.
>
> - As discussed in Section 2, there are already many outlier detection methods, ... ... Please see the paper:
> -- Zimek, A., Schubert, E., Kriegel, H.-P., A survey on unsupervised outlier detection in high-dimensional numerical data, Statistical Analysis and Data Mining (2012)
> ... ...
> recommend to add some distance-based outlier detection methods as baselines in experiments.
>
> In the second paragraph in section introduction, we point out several names for the task setting described in our paper, namely, outlier detection, Novelty Detection Concept Learning and One-class classification. They are used interchangeably in our paper. However, they may have specific meaning in other works.
>
> The distance-based method described in the above paper is an outlier detection method. It aims to find the samples that different from most other samples. Our paper focuses on novelty detection, which tries to find samples different from the given positive samples. When a large mount similar outliers exist in the task,  those two methods given two different results. The first method does not work anymore. For instance, in our experimental setting, the outliers are similar (from the same class). The distance-based methods, such as DBSCAN, OPTICS and LOF will identifier the most all outliers as inliers.
>
> But you do make a good point here, we can adapt their approaches in our task setting (novelty detection), which may be an idea of a new paper in my opinion.
>
> - Since parameter tuning by cross-validation cannot be used due to missing information of outliers, ... ... how to set these parameters to get better results is not obvious.
>
> Generally, in OCC task, we have no available outliers to do cross-validation. In our paper, we propose a measure to select models, which does not require outliers in validation dataset. It is called positively biased AUC score (see Figure 3 in the paper). The selected model by this measure is not necessarily optimal, but near optimal.

---

### Official Review · AnonReviewer1 · 2017-11-27
**In my view this paper is a clear rejection, a few interesting heuristics are presented without sufficient theoretical or experimental justification**

**Rating:** 3
**Confidence:** 5

**Review:**

This paper addresses the problem of one class classification. The authors suggest a few techniques to learn how to classify samples as negative (out of class) based on tweaking the GAN learning process to explore large areas of the input space which are out of the objective class.

The suggested techniques are nice and show promising results. But I feel a lot can still be done to justify them, even just one of them. For instance, the authors manipulate the objective of G using a new parameter alpha_new and divide heuristically the range of its values. But, in the experimental section results are shown only for a  single value, alpha_new=0.9 The authors also suggest early stopping but again (as far as I understand) only a single value for the number of iterations was tested.

The writing of the paper is also very unclear, with several repetitions and many typos e.g.:

'we first introduce you a'
'architexture'
'future work remain to'
'it self'

I believe there is a lot of potential in the approach(es) presented in the paper. In my view a much stronger experimental section together with a clearer presentation and discussion could overcome the lack of theoretical discussion.

---

> ### Author Response · Authors · 2017-12-16
> **RE: a few interesting heuristics are presented without sufficient theoretical or experimental justification**
>
> Thank you for your comments and the useful suggestions. Our response to your questions is as follows:
>
> 1) As you point out, we just show the experiment result for a single value alpha_new=0.9. Actually, we justify the choice in the Table 1.
>
> alpha_new = 1 → The objective of Generator is the exact same as in the original GAN. See Equation 1 in the paper:
> "Goodfellow, Ian, et al. "Generative adversarial nets." Advances in neural information processing systems. 2014."
>
> alpha_new ∈ (∼ 0.9, 1) → The adjustment of the objective of Generator is proposed to improve the training process of GANs. See section  3.4 One-sided label smoothing in the paper:
> "Salimans T, Goodfellow I, Zaremba W, et al. Improved techniques for training GANs[C]//Advances in Neural Information Processing Systems. 2016: 2234-2242."
>
> In those two cases, the training process will converge, which is what we want to avoid.
>
> alpha_new ∈ (0, ∼ 0.5) → The Generator has similar objective as the Discriminator. It will tend to generate data, from which the Discriminator is able to distinguish training data. That is to say that all the generated data distribute far from the positive class.
>
> What we want is that the Generator is capable of generating outliers that explore as a large place as possible (both the place far from and around the positive class), especially the place near the positive class. The values in the interval  (∼ 0.5, ∼ 0.9) are good candidates. If we assign one value from this interval to alpha_new, the Generator will generate data far from positive class at the beginning of the training phase because of the random initialization. After several training epochs, it will generate data that distribute around the positive class.
>
> Since we aim to build a robust outlier detector against any kind of outliers (including the ones distribute near the positive class), we choose 0.9 for the Generator so that the generated outliers can cover more space around the positive class to form a tight boundary.
>
> That is how the choice is justified. The above explanations are not only our intuitive understanding. They are also supported by experiments. Since the corresponding experiments are not the core part of our paper, we do not incorporate them in the experiment section in our paper.
>
> What is more, we may need to clarify, why not 0.91 or 0.89. The same problem in the paper [Improved techniques for training GANs]: why not take the value in the interval (0.91, 1) or (0.89, 1)to improve the training process of GAN? We need more details about the theoretical foundation, which is mentioned in our future work.
>
> 2) Early stopping is proposed to avoid the convergence. There are two ways to implement Early Stopping:
>    a): Stop the training process at a certain epoch, as you understand.
>    b): As we described in the section 4.1 Early stopping in our paper, we do not stop the training at a certain epoch. We just save the best model we get, with the training process going on. At the end, we take the most recent saved model as the final model. It is similar to the model selection. The measure we use to select model is the score (positive biased AUC), which does not require negative samples in the validation dataset. The measure is proposed in our paper (see Figure 3).
>
> 3) You are right, we definitely should fix the typos and correct the expression of some sentences.
>
> Looking forward to a further discussion with you!

---

### Author Response · Authors · 2018-01-04
**Revision Reminder**

Thank you again for the review comments. We take some useful suggestions from them and revise our paper in the following aspects:
1. formulate and express our proposed improved techniques mathematically
2. clarify the justifications of each proposed technique
3. provide more evaluation measures (ROC and AUC) and a stronger discussion about experiment results
4. correct the typos and polish some expressions

---

### Decision · Program_Chairs · 2018-01-29
**ICLR 2018 Conference Acceptance Decision**

**Decision:**

Reject

**Comment:**

This paper presents a framework where GANs are used to improve detection of outliers (in this context, instances of the “background class”). This is a very interesting and, as demonstrated, promising idea. However, the general feeling of the reviewers is that more work is needed to make the technical and evaluations parts convincing. Suggestions for further work towards this direction include: theoretical analysis, better presentation of the manuscript and, most importantly, stronger experimental section.